# G Protein-Dependent Activation of the PKA-Erk1/2 Pathway by the Striatal Dopamine D1/D3 Receptor Heteromer Involves Beta-Arrestin and the Tyrosine Phosphatase Shp-2

**DOI:** 10.3390/biom13030473

**Published:** 2023-03-03

**Authors:** Federica Bono, Zaira Tomasoni, Veronica Mutti, Giulia Sbrini, Rajesh Kumar, Francesca Longhena, Chiara Fiorentini, Cristina Missale

**Affiliations:** 1Section of Pharmacology, Department of Molecular and Translational Medicine, University of Brescia, Viale Europa 11, 25123 Brescia, Italy; 2Seattle Children’s Research Institute, 1920 Terry Ave., Seattle, WA 98101, USA

**Keywords:** G protein-coupled receptors (GPCRs), heterodimerization, dopamine receptors, striatum, L-DOPA-induced dyskinesia (LID), tyrosine phosphatase Shp-2, beta-arrestin

## Abstract

The heteromer composed of dopamine D1 and D3 receptors (D1R–D3R) has been defined as a structure able to trigger Erk1/2 and Akt signaling in a G protein-independent, beta-arrestin 1-dependent way that is physiologically expressed in the ventral striatum and is likely involved in the control of locomotor activity. Indeed, abnormal levels of D1R-D3R heteromer in the dorsal striatum have been correlated with the development of L-DOPA-induced dyskinesia (LID) in Parkinson’s disease patients, a motor complication associated with striatal D1R signaling, thus requiring Gs protein and PKA activity to activate Erk1/2. Therefore, to clarify the role of the D1R/D3R heteromer in LID, we investigated the signaling pathway induced by the heteromer using transfected cells and primary mouse striatal neurons. Collectively, we found that in both the cell models, D1R/D3R heteromer-induced activation of Erk1/2 exclusively required the D1R molecular effectors, such as Gs protein and PKA, with the contribution of the phosphatase Shp-2 and beta-arrestins, indicating that heterodimerization with the D3R abolishes the specific D3R-mediated signaling but strongly allows D1R signals. Therefore, while in physiological conditions the D1R/D3R heteromer could represent a mechanism that strengthens the D1R activity, its pathological expression may contribute to the abnormal PKA-Shp-2-Erk1/2 pathway connected with LID.

## 1. Introduction

The D1 (D1R) and D3 (D3R) receptors for dopamine (DA) possess peculiar functional and biochemical characteristics [1,2]. The D1R is the most widespread post-synaptic DA receptor in the brain and is typically coupled to Galfas/olf proteins leading to cAMP production and protein kinase A (PKA) activation [1,2]. The D1R has also been associated with the activation of the extracellular signal-regulated kinase 1/2 (Erk1/2) cascade by a mechanism depending on the cAMP/PKA pathway and involving both the DA- and cAMP-regulated phospho-protein-32 (DARPP-32) [3] and the tyrosine phosphatase Shp-2 [4,5]. Among the DA receptors, D1R possesses the lowest affinity for DA [6]. By contrast, post-synaptic D3R shows a restricted pattern in its distribution that includes the striatum, with the ventral part showing higher expression levels than the dorsal one [7,8,9,10]. Stimulation of the D3R, which is characterized by the highest affinity for DA [6], triggers several intracellular signaling pathways, including the inhibition of PKA [1] and the activation of Akt and Erk1/2 via recruitment of Gi proteins and stimulation of phosphatidylinositol-3 kinase (PI3K) [11,12,13,14,15].

D1R and D3R share the ability to form heteromers with other GPCR or non-GPCR. Interestingly, a heteromeric complex formed by these receptors (D1R/D3R) [16,17,18] has been identified in the GABAergic neurons of the striatum where D1R and D3R are co-localized and functionally correlated [19,20,21,22,23,24]. Some distinguishing features of the D1R/D3R heteromer have been defined using both transfected cells and experimental animal models [16,17,18,25]. In particular, BRET and bimolecular fluorescence complementation (BiFC) in combination with analysis of cell signaling suggested that the D1R/D3R heteromer is arranged as a quaternary structure made of two different D1R and D3R homodimers, each able to recruit its preferred G protein [18]. Using transfected cells, a synergistic interaction has been demonstrated in which heteromerization with the D3R results in increased D1R agonist affinity [16]. Moreover, the simultaneous stimulation of D1R and D3R within the heteromer results in the synergistic activation of Erk1/2 and Akt [18,25] in a G protein-independent, beta-arrestin 1-dependent way [18]. In addition, in the ventral part of the striatum, stimulating the D1R/D3R heteromer led to the selective activation of Akt but not of Erk1/2, an effect underlying the locomotor synergistic effects of D1R and D3R agonists [25]. In the dorsal part of the striatum, the D1R/D3R heteromer has been associated with the development of L-DOPA-induced dyskinesia (LID), a severe and irreversible motor complication related to the long-term administration of L-DOPA in Parkinson’s disease (PD) [24,26,27]. A growing body of evidence points to striatal D1R abnormal signaling, implying aberrant activation of Gs protein-induced cAMP accumulation, PKA activation, and Erk1/2 phosphorylation as the main molecular marker associated with LID development [28]. Our observation in LID models that aberrant activation of Shp-2, a D1R-interacting tyrosine phosphatase, potentiates D1R-mediated activation of Erk1/2 in striatal neurons [4,5,29] and also confirms the pivotal role of D1R signaling in LID. Of relevance, striatal D3R and D1R/D3R heteromer levels are abnormally elevated in LID, as observed in both animal models [20,30] and PD patients [31,32]. The molecular mechanisms associating the abnormal activity of the D1R/D3R heteromer with the PKA-Erk1/2 pathway and LID development, however, still need to be clarified.

Therefore, in this study, transfected cells and striatal neurons were used to further investigate the D1R/D3R heteromer signaling pathways focusing on the possible contribution of specific effectors, such as PKA and Shp-2, in activating the Erk1/2 pathway that classically represents the molecular event underlying LID.

## 2. Materials and Methods

### 2.1. Materials

Human embryonic kidney (Hek) 293 cells were provided by Open Biosystems (Shanghai, China). Tissue culture medium and fetal bovine serum (FBS) were purchased from Euroclone (Milan, Italy). The following compounds were purchased from Tocris (Bristol, UK): the selective D1R full agonist (±)-6-Chloro-2,3,4,5-tetrahydro-1-phenyl-1H-3-benzazepine hydrobromide (SKF81297), the selective D2-like agonist (4aR-trans)-4,4a,5,6,7,8,8a,9-Octahydro-5-propyl-1H-pyrazolo [3,4-g]quinoline hydrochloride (quinpirole), the selective D1-like antagonist (R)-(+)-7-Chloro-8-hydroxy-3-methyl-1-phenyl-2,3,4,5-tetrahydro-1H-3-benzazepine hydrochloride (SCH23390), the selective D2-like antagonist (RS)-(±)-5-Aminosulfonyl-*N*-[(1-ethyl-2-pyrrolidinyl)methyl]-2-methoxybenzamide (sulpiride), the protein kinase A inhibitor *N*-[2-[[3-(4-Bromophenyl)-2-propenyl]amino]ethyl]-5-isoquinolinesulfonamide dihydrochloride (H89), the prototypical PI 3-kinase inhibitor 2-(4-Morpholinyl)-8-phenyl-4H-1-benzopyran-4-one hydrochloride (LY294002) and the Gi/o protein inhibitor pertussis toxin (PTX). The -3,4-Dihydroxyphenethylamine hydrochloride (Dopamine), the monoclonal anti-alpha-tubulin antibody, and the monoclonal anti-D1R antibody were purchased from Sigma Aldrich (St. Louis, MO, USA). The anti-D3R antibody, anti-phospho-Erk1/2 antibody, anti-Erk1/2 antibody, anti-Shp2 antibody and the horseradish per oxidase (HRP)-conjugated secondary antibodies were from Santa Cruz (Santa Cruz Biotechnology Inc., Heidelberg, Germany). The anti-phospho-Akt (Thr308) antibody, the beta Arrestin 1 antibody and the beta-Arrestin2 antibody were from Cell Signaling Technology (Beverly, MA, USA). The anti-GAD67 antibody was purchased from Merk Millipore. DyLightTM 488-conjugated secondary antibodies were from Jackson Immuno Research Inc. (West Grove, PA, USA). Human D1R/pcDNA3.1 vector was kindly provided by Dr. Marc Caron (Duke University, Durham, NC, USA); GFP-DRD3 was a gift from Jean-Michel Arrang (Addgene plasmid # 24098; http://n2t.net/addgene:24098 (accessed on 12 September 2017); RRID: Addgene_24098) [33]; human mutant Shp-2 (C/S)/pcDNA3 vector was kindly provided by Dr. C. Nahmias (Institute Cochin, INSERM Paris, France). The small interfering RNA (siRNA) targeting beta-Arrestin1 and beta-Arrestin2 were from Sigma Aldrich (St. Louis, MO, USA). Cell permeable interfering TAT peptides TAT-D3R (NH2-YGRKKRRQRRRLKQRRRKRIL-COOH) and TAT-D3R-Sc scrambled sequence (NH2-YGRKKRRQRRRIRKLRLRQRK-COOH) were purchased from GenScript (Piscataway, NJ, USA) [34].

### 2.2. Animals

Mice genetically deprived of D3R (D3R-KO) were obtained from the Jackson Laboratory (B6.129S4-Drd3Tm1Dac/J) [35]. Homozygous mice were generated on a pure genetic background and were compared to syngeneic wild-type mice. The genotype of D3R-KO mice was determined as described by Accili et al. [35]. Mice breeding was performed to achieve timed pregnancy with an accuracy of +0.5 days. The embryonic day (E) was determined by considering the day of insemination (determined using a vaginal plug) as day E 0.5. Animal care was in accordance with the European Community Council Directive, September 2010 (2010/63/EU) with the approval of the Institutional Animal Care and Use Committee of the University of Brescia, and in line with Italian law.

### 2.3. Cell Cultures, Transfection and Treatments

Hek293 cells were cultured in Dulbecco’s modified Eagle’s medium containing 10% fetal bovine serum (FBS), 2 mM Glutamine, 0,1 mM non-essential amino acids (NEAA), 100 U/mL penicillin and 100 μg/mL streptomycin (P/S). Hek293 cells individually expressing D1R (Hek-D1R) or D3R (Hek-D3R) or co-expressing D1R and D3R (Hek-D1RD3R) were generated using transient transfection with the D1R/pcDNA2.1 (1 μg), the GFP-DRD3 (1 μg) or both plasmids (0.5 μg for each) using the Arrest-IN reagent (Thermo Scientific) according to the manufacturer’s instructions. The pCDNA3 vector was used to equilibrate the total amount (1 μg) of transfected DNA. After 16 h (h) of serum starvation, cells were treated with DA receptor agonists (Dopamine; SKF81297, quinpirole) and antagonists (SCH23390, sulpiride) as well as with inhibitors of specific signaling pathways (H89, LY294002, PTX), at the concentrations and for the times indicated in the text and in the figure legends. In a set of experiments, Hek293 cells expressing D1R or co-expressing D1R and D3R were transiently transfected with Shp-2 (C/S)/pcDNA3 vector (1 μg total) using the Arrest-IN reagent and treated as described in the text and in the figure legends. Moreover, treatments of Hek293 cells co-expressing D1R and D3R were performed in the presence or in the absence of the interfering peptides TAT-D3R (1 μM) or TAT-D3R-sc (1 μM). Cells were analyzed for Erk1/2 phosphorylation (p-Erk1/2) and Akt phosphorylation (p-Akt) using Western Blot (WB), as described below.

### 2.4. Primary Mouse Striatal Neuron Cultures and Treatments

Primary striatal neurons were prepared from 16.5 days mouse embryos (WT or D3R-KO). The striatum was removed and mechanically dissociated, and the cells were suspended in Neurobasal Medium (Gibco-Invitrogen, Carlsbad, CA, USA), added with B27 supplement (Gibco-Invitrogen) and 2 mM glutamine. Cells were seeded on poly-D-Lysine-coated dishes and cultured at 37 °C for 7 days (div 7), in a humidified atmosphere of 5% CO2 and 95% air. Striatal neurons from WT mice were incubated in the absence of serum for 16 h and treated with SKF81297 (10 μM) or quinpirole (10 μM) for 5 min, in the presence or in the absence of H89 (1 μM) and analyzed for p-Erk1/2 and p-Akt with WB, as described below. In another set of experiments, striatal neurons from the WT and D3R-KO mice were treated with SKF81297 at different concentrations (0.001–10 μM) for 5 min and analyzed for p-Erk1/2 with WB. Moreover, striatal neurons were exposed to TAT-D3R (1 μM) or TAT-D3R-sc (1 μM) peptides for 1h and analyzed using a proximity ligation assay (PLA) as described below.

### 2.5. Beta-Arrestin 1 and Beta-Arrestin 2 Gene Silencing in Primary Striatal Neurons Using RNA Interference

Two specific siRNA duplexes targeting different regions of the mouse beta-arrestin 1 (si-bArr1a: SASI_Mm01_00146723; si-bArr1b: SASI_Mm01_00146722) or beta-arrestin 2 gene (si-bArr2a: SASI_Mm02_00340782; si-bArr2b: SASI_Mm01_00076524) and a non-targeting scrambled control siRNA duplex were used (Sigma Aldrich, St. Louis, MO, USA). Primary striatal neurons (div 7) were transfected with different concentrations (25–50 nM) of each beta-arrestin1/ beta-arrestin2 siRNAs or the scrambled siRNA using the X-tremeGENE siRNA transfection reagent (Roche, Mannheim, Germany). The efficiency of gene silencing was evaluated 120 h post-transfection by measuring beta-arrestin1 and beta-arrestin2 protein levels with WB. The si-bArr1b (50 nM) and the si-bArr2a (50 nM) were the most effective and selective siRNA producing about ~0% and about 70% inhibition of beta-arrestin1 and beta-arrestin2 expression, respectively, and were used in subsequent experiments. Striatal neurons were transfected with a combination of si-bArr1b (50 nM) and si-bArr2a (50 nM) or the scrambled siRNA (50 nM) for 120 h, and the interaction of Shp-2 with the D1R was analyzed using co-immunoprecipitation, as described below. Moreover, striatal neurons transfected with si-bArr1b/si-bArr2a (both at 50 nM) or the scrambled siRNA (50nM), were treated with SKF81297 (10μM) or quinpirole (10 μM) for 5 min and analyzed for p-Erk1/2 with WB, as described below.

### 2.6. Protein Preparation, Electrophoresis and Western Blot (WB)

Cells were lysed in ice-cold RIPA buffer [50 mM Tris-HCl pH 7.5, 150 mM NaCl, 1% NP40, 0.1% sodium dodecyl sulfate (SDS) and 0.5% sodium deoxycolate], containing 2 mM Na_3_VO_4_, 10 mM NaF and a complete set of protease inhibitors (Roche Diagnostics, Mannheim, Germany). The protein concentration was determined using the DC Protein Assay Reagent (Bio-Rad, Milano, Italy). Aliquots of total proteins were resolved with SDS-polyacrilamide gel electrophoresis, transferred onto polyvinylidene fluoride (PVDF) membrane and blotted for 1 h at room temperature (RT) in Tris-buffer saline containing 0.1% Tween-20 and 5% non-fat powdered milk. Membranes were incubated overnight at 4 °C with primary antibodies anti-p-Erk1/2 (1:500), anti p-Akt (1:1000), anti-beta-arrestin1 (1:1000), anti-beta-arrestin2 (1:1000), anti-Shp2 (1:1000) or anti total Erk1/2 (1:1000). Detection was performed using chemiluminescence with horseradish peroxidase-conjugate secondary antibodies (1:1500). Membranes were stripped and re-probed for loading with anti-alfa-tubulin (1:300,000). Densitometric analysis of the immunoblots was performed using the GelPro-Analyzer (Media Cybernetics, Bethesda, MD, USA).

### 2.7. Co-Immunoprecipitation and WB

Cells were lysed in ice-cold RIPA buffer containing 2 mM Na_3_VO_4_, 10 mM NaF and a complete set of protease inhibitors (Roche Diagnostics, Mannheim, Germany). The protein concentration was determined using the DC Protein Assay Reagent (Bio-Rad, Milano, Italy). Protein aliquots (60 μg for each condition) were incubated overnight at 4 °C with the anti-D1R antibody (1:40 dilution) in RIA buffer (400 mM NaCl, 20 mM EDTA, 20 mM Na_2_HPO_4_) containing 0.1% SDS. Protein-A agarose beads (Santa Cruz Biotechnology Inc., Heidelberg, Germany) were added, and incubation was continued for 3 h at RT. The beads were collected and extensively washed with a buffer containing 1% NP-40, 10% sodium dodecyl sulfate (SDS), 5% sodium deoxycolate and 10 μg/mL phenylmethylsulfonyl fluoride (PMSF) in PBS. The resulting proteins were analyzed using WB with the anti-Shp2 (1:1000) antibody.

### 2.8. Proximity Ligation Assay (PLA)

The ‘‘in situ’’ PLA analysis was performed in Hek 293 cells co-expressing D1R and D3R, in mouse brain sections and in primary striatal neuron cultures using the Duolink in situ reagents (O-LINK Bioscience, Upsalla, Sweden), following the manufacturer directions. Mice were anesthetized with chloral hydrate (400 mg/kg ip) and perfused with 4% paraformaldehyde, and the brains were removed and cryoprotected with 20% sucrose. Coronal sections (30 μm thick) containing the striatum were cut and mounted on microscope slides. Striatal brain sections or fixed cells were incubated with blocking solution for 30 min at 37 °C and then with a rat anti-D1R (1:100; Sigma Aldrich, St Louis, MO, USA) and a goat anti-D3R (1:50; Santa Cruz Biotechnology, Heidelberg, Germany) primary antibodies overnight at 4 °C. Samples were then washed twice in Wash Buffer A (O-LINK), and incubated with the PLA probe solution (O-LINK), containing anti-goat PLA probe MINUS and anti-rat PLA probe PLUS, for 60 min at 37 °C. Samples were then washed in Wash Buffer A and incubated with the ligation solution, containing the DNA ligase which allows the ligation of the probes and two oligonucleotides, to form a round circle DNA strand, for 30 min at 37 °C. Then, samples were washed in Wash Buffer A and incubated with the amplification solution, consisting of fluorescently labeled oligonucleotides and the DNA polymerase for the rolling circle amplification, at 37 °C for 100 min. Samples were washed twice in 1× Wash Buffer B (O-LINK) and once in 0.01× Wash Buffer B. In experiments with striatal neurons, the samples were subsequently incubated with a mouse anti-GAD67 (1:500; Millipore, Burlington, MA, USA) primary antibody for 120 min at RT. Samples were washed twice in PBS-Triton 0,1% (TBS-T) and incubated with a DyLightTM 488-conjugated secondary antibody (Jackson Immunoresearch, West Grove, PA, USA) for 30 min at RT. Finally, cells were mounted using the Duolink In Situ Mounting Medium with DAPI (O-LINK, Uppsala, Sweden) and analyzed using a Zeiss LSM 510 Meta confocal microscope equipped with Plan-Apochromat 63x/1.4 numerical aperture oil objective and LSM 510 Meta Software, version 3.5 (Carl Zeiss AG, Oberkochen, Germany). PLA signals were quantified with NIH ImageJ software (version 2.1.0; Bethesda, MD, USA) after image deconvolution. The number of cells showing discrete positive red spots and the average number of PLA puncta per cell have been quantified from images acquired from at least ten randomly chosen fields per condition from three independent experiments.

### 2.9. Statistical Analysis

Each experiment was repeated at least three times and values are expressed as mean ± SEM. Significant differences were determined using analysis of variance (one-way ANOVA) followed by Bonferroni post-test versus untreated cells, provided by GraphPad PRISM 4 (GraphPad Software, San Diego, CA, USA).

## 3. Results

### 3.1. In Hek293 Cells Expressing the D1R/D3R Heteromer, the Simultaneous Stimulation of D1R and D3R Selectively Activates the PKA/Erk1/2 Pathway

We used Hek-D1R cells and Hek-D3R cells to define the ability of D1R or D3R in inducing Erk1/2 (p-Erk1/2) and Akt (p-Akt) phosphorylation. The PKA inhibitor H89, the PI3K inhibitor LY294002 and the Gi/o inhibitor PTX were used to investigate the mechanisms involved in the activation of Erk1/2 or Akt. Inhibitors were used at concentrations that did not change the baseline levels of these cascades in untreated cells (Appendix A). In line with previous data [11,12,36], D1R stimulation induced the transient phosphorylation of Erk1/2, but not of Akt, in a PKA-dependent manner (Appendix A). As previously reported [37,38,39,40,41], D3R stimulation induced Erk1/2 and Akt transient phosphorylation by a Gi-dependent activation of PI3K (Appendix A).

The ability of the D1R/D3R heteromer in modulating Erk1/2 and Akt signaling was analyzed in HekD1R/D3R cells. As previously reported, co-transfection of D1R and D3R in Hek293 cells results in a high level of membrane receptor co-localization that is best explained by the formation of the D1R/D3R heteromer [16]. In situ PLA was used to detect the D1R/D3R heteromer in these cells. As shown in Figure 1, an intense PLA signal with a dotted appearance was detected in Hek-D1R/D3R cells suggesting close proximity between the D1R and the D3R. The PLA signal was quantified by counting the number of cells with red spots normalized over the total amount of cells counterstained with DAPI. Moreover, the PLA spots for cells were automatically determined using the NIH ImageJ software (version 2.1.0; Bethesda, MD, USA). Positive spots were observed in approximately 80% of D1R/D3R transfected cells, each having 22.3 red spots/cell (Figure 1, panel a). PLA was also performed in the presence of DR3-related interfering peptides and their scrambled counterparts [34,42,43] (both at 1 µM) for 30 min. As shown in Figure 1 (panels d–f), the PLA signal was significantly affected by pretreatment with TAT-D3R. The positive red spots were in fact observed in less than 1% of cells, each having 1,2 red spots/cell. By contrast, TAT-D3R-sc (1 µM; 30 min) did not significantly affect the PLA signal. Positive red spots were in fact observed in 75% of cells, each having 20.8 red spots/cell (panels g–i).

Hek-D1R/D3R cells were then treated with the D1R agonist SKF81297 in combination with the D2R/D3R agonist quinpirole (both 1 μM) for 5, 15 or 30 min (Figure 2A). Analysis of p-Erk1/2 levels showed that D1R and D3R co-stimulation transiently promoted Erk1/2 phosphorylation, reaching a peak at 5 min of stimulation; by contrast, the same treatment did not increase p-Akt levels (Figure 2A,C). Moreover, in Hek-D1R/D3R cells, Erk1/2 phosphorylation induced with SKF81297/quinpirole (both at 1μM; 5 min) was specifically prevented by H89 (1 µM) (Figure 2D,E), while pre-incubation with LY294002 (10 µM) or with PTX (100 ng/mL) did not modify SKF81297/quinpirole-induced Erk1/2 phosphorylation (Figure 2D–F). Similar results were obtained by stimulating Hek-D1R/D3R cells with dopamine (DA) (1 μM). Indeed, analysis of p-Erk1/2 and p-Akt levels showed that DA transiently promoted Erk1/2 (peak at 5 min) but not Akt phosphorylation (Figure 2G–I); moreover, Erk1/2 phosphorylation induced by DA was specifically prevented by H89 (1 µM) but not by LY294002 (10 μM) or by PTX (100 ng/mL) (Figure 2J–L). Therefore, in Hek-D1R/D3R cells, co-stimulating D1R and D3R results in the selective activation of the Erk1/2 cascade, an effect requiring PKA, thus resembling the intracellular signaling activated by the D1R when individually expressed.

### 3.2. In Hek293 Cells Expressing the D1R/D3R Heteromer, the Individual Stimulation of D1R and D3R Specifically Activates the PKA/Erk1/2 Pathway

Hek-D1R/D3R cells were incubated with either SKF81297 (1 μM; 5 min) or quinpirole (1 μM; 5 min) in the presence or in the absence of H89 (1 µM), LY294002 (10 µM) or PTX (100 ng/mL) and analyzed for p-Erk1/2 and p-Akt levels (Figure 3). The results show that D1R stimulation significantly induced Erk1/2, but not Akt, phosphorylation (Figure 3B,C), an effect that was blocked by H89 but not by LY294002 and PTX (100 ng/mL) (Figure 3B). Interestingly, the stimulation of D3R with quinpirole also resulted in Erk1/2, but not Akt, phosphorylation, an effect prevented by H89 but not by LY294002 and PTX (Figure 3D–F). Exposure of Hek-D1R/D3R cells to TAT-D3R (1 µM; 30 min), but not toTAT-D3R-sc (1 µM; 30 min), restored the capability of quinpirole (1 µM; 5 min) to activate Akt, providing evidence that within the heteromer, D3R loses the property to activate its typical intracellular signaling (Figure 3G,H). Thus, in Hek293 cells expressing the D1R/D3R heteromer, both the individual stimulation and the co-stimulation of the two interacting receptors results in the selective activation of the PKA-Erk1/2 pathway, associated with the D1R protomer, as a result of allosteric interactions occurring within the heteromer. Such allosteric interactions are also involved in the cross-antagonism at the Erk1/2 level. Antagonists negatively regulate, in fact, the functional properties of the receptor on which they do not bind directly. This property, which was also reported by Guitart et al. [18], was demonstrated in Hek-D1R/D3R cells treated with DA or with either D1R or D2R/D3R agonists in the presence of either SCH 23390 or sulpiride (Appendix A).

### 3.3. In Hek-D1R/D3R Cells, D3R Interaction with D1R Increases the Ability of DA and D1R Agonists in Activating the ERK1/2 Pathway

We previously demonstrated that in Hek-D1R/D3R cells, D1R displays a higher affinity for DA due to its interaction with the D3R, leading to increased potency in stimulating cyclic AMP (cAMP) formation [16]. Since in Hek-D1R/D3R cells, activation of Erk1/2 by D1R and D3R stimulation depends on the cAMP/PKA pathway, we investigated whether DA potency in activating the Erk1/2 cascade would be equally increased. For this aim, Hek-D1R and Hek-D1R/D3R cells were treated with DA (1–10 µM) for 5 min and analyzed for p-Erk1/2. As shown in Figure 4, in HekD1R cells, DA dose-dependently increased pErk1/2 levels, with significant values starting at 100 nM and reaching a maximum at 1 μM (Figure 4A,B). Interestingly, in Hek-D1R/D3R cells, DA-induced Erk1/2 phosphorylation was already maximal at 1 nM (Figure 4A,C). To investigate the role of D3R in increasing D1R-mediated Erk1/2 activation, Hek-D1R and Hek-D1R/D3R cells were treated with SKF81297 (1–10 µM) for 5 min and analyzed for p-Erk1/2. As shown in Figure 4, in cells expressing D1R, p-Erk1/2 levels increased in a dose-dependent manner, with significant values starting at 100 nM and reaching the maximum at 1 μM (Figure 4D,E), while in Hek-D1R/D3R cells, the effect of SKF81297 was maximal at 1 nM (Figure 4D,F). Together, these data suggest that heteromerization with the D3R is by itself sufficient to increase the affinity of D1R for agonists in activating the Erk1/2 cascade.

### 3.4. In Hek-D1R/D3R Cells, Activation of Erk1/2 Induced with D1R and D3R Co-Stimulation Requires the Tyrosine Phosphatase Shp-2

We previously reported that D1R-mediated activation of Erk1/2 requires PKA that, in turn, modulates the activity of Shp-2, a tyrosine phosphatase associated with the D1R and mediating D1R-induced Erk1/2 activation [4,5,29]. Since our data have shown that stimulation of the D1R/D3R heteromer results in the activation of PKA/Erk1/2 pathway, we investigated the role of Shp-2 in our cell model. Hek-D1R and Hek-D1R/D3R cells were transfected with a dominant-negative form of Shp-2, Shp-2(C/S) [4,44], and treated with DA (1 µM; 5 min). As shown in Figure 5, in both Hek-D1R and Hek-D1R/D3R cells, the expression of the inactive Shp-2(C/S) abolished DA-induced Erk1/2 activation, suggesting that D1R/D3R heteromer-induced activation of Erk1/2 requires Shp-2.

### 3.5. In Primary Cultures of Striatal Neurons, Activation of the Erk1/2 Pathway with Either D1R or D3R Requires PKA

The existence of the D1R/D3R heteromer in striatal neurons was investigated in brain sections, containing the striatum from both wild-type and D3R knock-out mice (D3R-KO) using PLA. As shown in Figure 6A, positive red spots were observed in approximately 45% of wt cells (each having 13.5 red spots/cell) (panel a–c), but not in D3R-KO cells (panel d–f). Similarly, we used PLA to visualize the D1R/D3R heteromer in primary cultures of striatal neurons prepared from 16.5-day mouse embryos, as previously described [23]. Neurons were treated for 30 min with TAT-D3R or TAT-D3Rsc (both at 1 µM). Analysis of PLA signal showed that red spots were detectable in approximately 25% of GAD67-positive neurons, each having 30,4 red spots/cell, of both untreated (Figure 6B, panel a–c) and TAT-D3Rsc-treated cultures (Figure 6B, panel g–i); by contrast, the PLA signal was significantly reduced in TAT-D3R-treated neurons, with positive red spots detectable in less than 5% of GAD67-positive neurons, each having 4,3 red spots/cell (Figure 6A, panel d–f). Overall, these data suggest that striatal GABA-ergic neurons express a significant level of D1R/D3R heteromer and, therefore, may represent a suitable physiological model to investigate the properties of this receptor complex.

Striatal cultures were treated with either SKF81297 (10 µM) or quinpirole (10 µM) for 5 min in the presence or in the absence of H89 (1 µM) and analyzed for Erk1/2 and Akt phosphorylation. As shown in Figure 6C,D, stimulation with SKF81297 significantly increased p-Erk1/2 levels, an effect that was prevented by H89; by contrast, SKF81297 failed to induce Akt phosphorylation (Figure 6E). Similarly, quinpirole stimulation resulted in the phosphorylation of Erk1/2 (Figure 6F,G), but not of Akt (Figure 6H), an effect that was prevented by PKA inhibition. Therefore, as shown in transfected Hek293 cells and also in striatal neurons, where the D1R/D3R heteromer is endogenously expressed, both D1R and D3R agonists promote PKA-dependent Erk1/2 phosphorylation but not Akt phosphorylation. These data thus confirm that in striatal neurons, interaction with D1R strongly impacts the ability of D3R in activating its typical intracellular signaling.

### 3.6. In Primary Striatal Neurons, D3R Dimerization with D1R Increases the Ability of D1R Agonists in Activating the ERK1/2 Pathway

The ability of D3R in increasing D1R affinity for agonists was next investigated in mouse primary striatal neurons, derived from both wild-type and D3R-KO mice. For this aim, striatal neurons were treated with SKF81297 at different concentrations (1–10 µM) for 5 min and analyzed for p-Erk1/2 levels. As shown in Figure 7, in neurons from D3R-KO mice expressing only D1R, we observed a dose-dependent phosphorylation of Erk1/2, which is significant with SKF81297 at 100 nM (Figure 7A,B). By contrast, in neurons from wild-type mice co-expressing D1R and D3R, SKF81297 significantly increased p-Erk1/2 at 10 nM, with steady levels of phosphorylation despite the increase in SKF81297 concentration (Figure 7A,C). Therefore, in striatal neurons, the D1R ability to activate the Erk1/2 pathway is increased by dimerization with the D3R.

### 3.7. In Primary Striatal Neurons, Beta-Arrestin1 and Beta-Arrestin2 and Shp2 Are Required for D1R-D3R Heteromer-Induced Activation of PKA-Erk1/2 Signaling

Using heterologous systems, the recruitment of the adaptor protein beta-arrestin-1 has been described as an essential step for D1R/D3R heteromer to activate Erk1/2 [18]. We also found that D1R-D3R heteromer activation of the G-dependent/Erk1/2 cascade also requires Shp-2; interestingly, the requirement of beta-arrestin in the recruitment of Shp-1 and Shp-2 tyrosine phosphatases to their target receptors has been demonstrated [45,46,47]. Here, we analyzed the role of beta-arrestin1 and beta-arrestin2 in D1R/D3R heteromer-induced signaling using primary cultures of striatal neurons and the siRNA approach. A set of preliminary experiments has been performed in order to identify the most efficient and specific siRNA able to knock down endogenous beta-arrestin1 and beta-arrestin2, both expressed in striatal neurons. Of the two siRNAs tested, the si-bArr1b and the si-bArr2a were selected (Supplementary data; Appendix A). Striatal neurons were exposed to a combination of si-bArr1b and si-bArr2a (50 nM; 120 h) as well as to the scrambled siRNA, used as a control (si-scr; 50nM, 120 h). As shown in Figure 8A,B, si-bArr1b/si-bArr2a treatment, but not the scrambled one, remarkably reduced the neuronal expression of both beta-arrestin1 and beta-arrestin2; Shp-2 and Erk1/2 were also investigated showing that silencing beta-arrestin1 and beta-arrestin2 did not affect their expression (Figure 8B). Since previous works have suggested a crucial role of beta-arrestin for Shp-2 activity [45,46,47] lysates derived from si-bArr1b/si-bArr2a, si-scr-treated and untreated cells were used in co-immunoprecipitation experiments to investigate the interaction of Shp-2 with the D1R [4]. As shown in Figure 8C, incubation of striatal proteins derived from untreated neurons with the anti-D1R antibody immunoprecipitated a 70 kDa species specifically recognized by the anti-Shp-2 antibody, which was absent when the immunoprecipitating antibody was omitted. Interestingly, when co-immunoprecipitation experiments were carried out in lysates derived from si-bArr1b- and si-bArr2a-treated neurons, the 70 kDa corresponding to Shp-2 protein was absent, thus suggesting that D1R interaction with Shp-2 requires beta-arrestins. Then, si-b-Arr1b/b-Arr2a- or si-scr-treated and untreated neurons were incubated with SKF81297 (10 µM) or with quinpirole (10 µM) for 5 min and analyzed for p-Erk1/2 using WB. As shown in Figure 8D,E, both compounds induced robust phosphorylation of Erk1/2 in untreated and si-scr-treated neurons but not in b-Arr1b/b-Arr2a-treated neurons. Altogether, these data suggest that beta-arrestins, keeping Shp-2 together with the D1R, are crucially required for D1R-D3R heteromer ability in activating the PKA-Erk1/2 signaling.

## 4. Discussion

In this paper, we define the intracellular signaling activated by the D1R/D3R heteromer that was analyzed in transfected Hek293 cells [16,18,25] and primary cultures of mouse striatal neurons. The results show that both the coincident and the individual stimulation of D1R and D3R resulted in transient and PKA-dependent activation of the Erk1/2 cascade representing the typical D1R-related intracellular signaling. These observations suggest the existence of allosteric interactions between D1R and D3R as the molecular mechanisms underlying heteromer function [48,49]. As previously reported [18], both D1R and D3R antagonists counteract heteromer-mediated signaling evoked either by individual or coincident stimulation of interacting receptors. This mechanism known as “cross antagonism”, is another form of allosteric interaction observed for other GPCR heteromers [50,51].

The PKA-Erk1/2 pathway likely represents peculiar signaling activated by the D1R/D3R heteromer. Neither D1R/D3R co-stimulation nor the individual D3R stimulation, in fact, activated the Erk1/2 pathway through Gi protein and PI3K, as the D3R typically does when it is individually expressed [52]. Furthermore, D1R/D3R individual or co-stimulation did not promote Gi protein- and PI3K- dependent activation of Akt, a signaling pathway likely exclusive of D3R (Supplementary data). Intriguingly, the D3R ability in activating the Akt pathway was restored when the D1R/D3R heteromer was disrupted using specific interfering peptides, the most useful strategy to examine GPCR heteromer expression and function [42]. All these data strongly corroborate the idea that the D1R/D3R heteromer exclusively signals through a D1R-associated PKA-dependent mechanism, likely requiring the recruitment of the Gs protein. This observation is in contrast with the data reported by Guitart et al. in transfected cells [18]. However, our results were obtained in both transfected cells and cultured striatal neurons, a cell model widely used for investigating the molecular mechanisms regulating striatal DA receptor function. GABAergic neurons physiologically express in fact both D1R and D3R together with their typical partner proteins and effectors [53]. In these neurons, where D1R/D3R heteromers were identified with PLA, both D1R and D2R/D3R agonists activated the Erk1/2 pathway in a PKA-dependent way, while D2R/D3R agonists did not activate Akt. These data on one hand strongly confirm that the D1R/D3R heteromer exclusively signals through the PKA-Erk1/2 pathway. On the other hand, the observation that quinpirole preferentially activates PKA-Erk1/2 signaling suggests that the D1R/D3R heteromer is significantly expressed in these neurons. This conclusion is also supported by the observation that activation of both Erk1/2 and Akt in a Gi protein- and PI3K-dependent way, was undetectable in these cells. Therefore, the D1R/D3R heteromer represents a molecular entity with physiological significance. On this line, this heteromer has been detected with PLA in almost 50% of GABAergic neurons of rodent and monkey brain sections containing the dorsal part of the striatum [24, this work]. In addition, the fact that the results obtained in primary neuronal cultures are in line with those obtained in transfected cells suggests that our transfected cell system represents a suitable model to study the properties of the D1R/D3R heteromer. This is of relevance since it is well known that working with heterologous expression systems could originate artifacts according to the different levels of receptor expression. This is especially relevant when studying GPCR heteromers, as suggested by the study of opsin receptors [54]. Thus, in addition to the cellular context, receptor concentration directs the formation of a GPCR oligomer unit with multiple partners and distinct potential arrangements with proper signaling outputs [48,49,55,56]. These observations could explain the differences between our results and those reported by Guitart et al. [18,25].

Our previous data have shown that D1R/D3R heteromerization increases the affinity of DA for the D1R and the potency of DA in stimulating AC activity [16]. On this line, here we report that the D1R/D3R heteromerization significantly increased the ability of DA in activating the D1R-dependent PKA-Erk1/2 pathway. However, an intriguing observation is that in both transfected cells and primary neurons, D3R dimerization with D1R increased “per se” the D1R agonist efficacy in activating the PKA/Erk1/2 pathway. In other words, it is likely that the activity of D1R is allosterically regulated by the presence of the partner receptor, a mechanism that resembles that observed for orphan GPCRs [57,58,59]. It has been shown, in fact, that orphan GPCRs can heteromerize with GPCRs that have identified ligands and regulate their function. For instance, the orphan GPCR 143 has been described as a new interacting partner for D2R and D3R that negatively modulates their activity [60]. Similarly, the apo-ghrelin GHSR1a receptor, interacting with the D2R, allosterically modifies its canonical signaling [61]. On this line, D1R/D3R heteromerization could represent a molecular strategy by which the D3R amplifies D1R activity [16,17]. A similar mechanism has been described for the adenosine A1/A2A receptor heteromer [62,63,64]. Interestingly, this heteromer acts as a concentration-sensing device through the recruitment of both Gi and Gs proteins. In particular, at low adenosine concentrations, A1R-induced Gi-mediated signaling is predominantly activated, while at high adenosine concentrations, Gs protein is engaged via A2AR, with the long C-terminal domain of activated A2AR blunting Gi-mediated signaling [63]. Our data indicate that an asymmetric Gs protein-dependent signaling is always observed when the D1R/D3R heteromer is activated, regardless of doses and agonists; however, the mechanism by which D1R interaction with D3R prevents its coupling with the Gi protein/PI3K pathway remains to be investigated.

According to Guitart et al. [18,25], we found that beta-arrestins are crucial for D1R/D3R heteromer-induced PKA-Erk1/2 activation. It is well known that these proteins play multiple roles in GPCR function, from desensitization and internalization to regulation of multiple signaling pathways, related to their ability to interact with various effectors, including the tyrosine phosphatases Shp-1 and Shp-2 [65]. We have shown that Shp-2 is a partner protein for D1R in activating striatal PKA-Erk1/2 signaling, both in physiological conditions and in LID [4,5,29]. Thus, being the D1R/D3R heteromer involved in LID, the role of Shp-2 in its transductional activity was investigated. The data show that in transfected cells, Shp-2 is required for heteromer-induced activation of Erk1/2. Moreover, using primary striatal neurons and the RNA interference technique we found that reduced levels of beta-arrestin 1/2 significantly decreased the ability of D1R to interact with Shp-2, thus impairing D1R/D3R heteromer signaling. Hence, beta-arrestin1/2 are crucial for keeping Shp-2 together with the D1R. On this line, beta-arrestin 2 is fundamental for the recruitment of Shp-1/Shp-2 to the KIR2DL1 receptor on natural killer cells probably through direct binding with the receptor itself [46]. Moreover, a similar role of beta-arrestin 2 has been described for Shp-1 recruitment to the receptor for the bacterial adhesion molecule intimin that inhibits host innate immunity following microbial infection [47]. The exact role of each individual beta-arrestin and the molecular mechanisms involved in the interplay among the heteromer, Shp-2 and beta-arrestins have not been investigated yet. However, these data represent a further indication that the D1R/D3R heteromer signals through the transductional pathway typical of the D1R. Taken together, our data suggest that in the motor part of the striatum, the abnormal expression of D1R/D3R heteromer leading to PKA-Erk1/2 overactivation could represent the mechanism underlying D1R hyperactivation associated with LID [28]. Interestingly, our results, showing that D1R heteromerization with D3R is sufficient for increasing its activity, are in line with the notion that in animal models of PD, treatment with selective D1R agonists strongly induces dyskinetic behaviors [66].

To date, GPCR heteromerization with other receptors, including GPCR themselves, ionotropic and tyrosine kinase receptors appear as a widespread phenomenon allowing function and regulation of those receptors that are in close proximity within the plasma membranes of native tissues. In fact, while an essential prerequisite for heterodimerization is the simultaneous occurrence in the same cellular microdomain, the high conformational flexibility of GPCRs hypothetically permits engagement with a great variety of interacting partners [48,49]. On this line, the D3R is an interacting partner for several GPCRs [67], such as the D2R for DA [68], the A2A receptor for adenosine [69], the NTSR1 receptor for neurotensin [70,71] and the MT1 and MT2 receptors for melatonin [72]. Moreover, the interaction of the D3R with ionotropic receptors such as the alfa4-beta2 receptor for acetylcholine has been reported [15,34]. Similarly, the D1R is part of several heteromeric complexes, including those with the NMDA receptor for glutamate [73], the D2R [74,75,76] and the A1 and A2 adenosine receptors [77]. Interestingly, the interaction of D1R with the H3 receptor (H3R) for histamine, within the GABAergic neurons of the striatum, has been described. In this case, H3R ligands act as a “molecular brake” for D1R signaling [78,79,80]. Therefore, in addition to the notion that GPCR heteromers represent novel receptors with proper pharmacological, signaling and trafficking characteristics, heteromerization may also represent a molecular strategy to modulate the activity of a given receptor, as is the case of D1R with D3R and with H3R.

## 5. Conclusions

In conclusion, together with numerous other pieces of evidence, this study further points to the D1R/D3R heteromer as the molecular effector underlying the D1R-related dysfunction characterizing LID, thus indicating that targeting this heteromer could be a strategy for reducing the abnormal PKA-Erk1/2 striatal activation and LID behavior. Moreover, evidence that the D1R/D3R heteromer could also operate in physiological conditions to strengthen the activity of D1R when DA is low has also been provided.

## Figures and Tables

**Figure 1 biomolecules-13-00473-f001:**
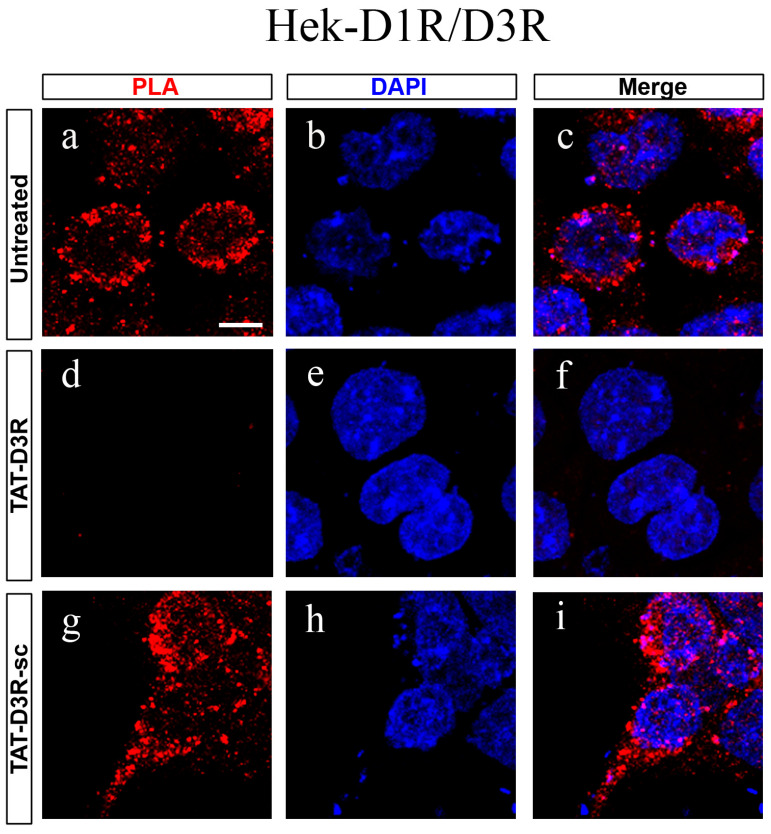
Detection of D1R/D3R heteromer with PLA in Hek293 cells expressing both D1R and D3R. Representative PLA image performed in Hek293 cells transiently expressing both D1R and D3R (Hek-D1R/D3R), untreated (panel **a**–**c**) and pre-treated with TAT-D3R (1 μM) (panel **d**–**f**) or TAT-D3R-sc (1 μM) (panel **g**–**i**) for 24 h (**h**); specific anti-D1R and anti-D3R antibodies were used. PLA signals appear as red spots; nuclei are detected with DAPI (blue). Scale bar = 20 μm.

**Figure 2 biomolecules-13-00473-f002:**
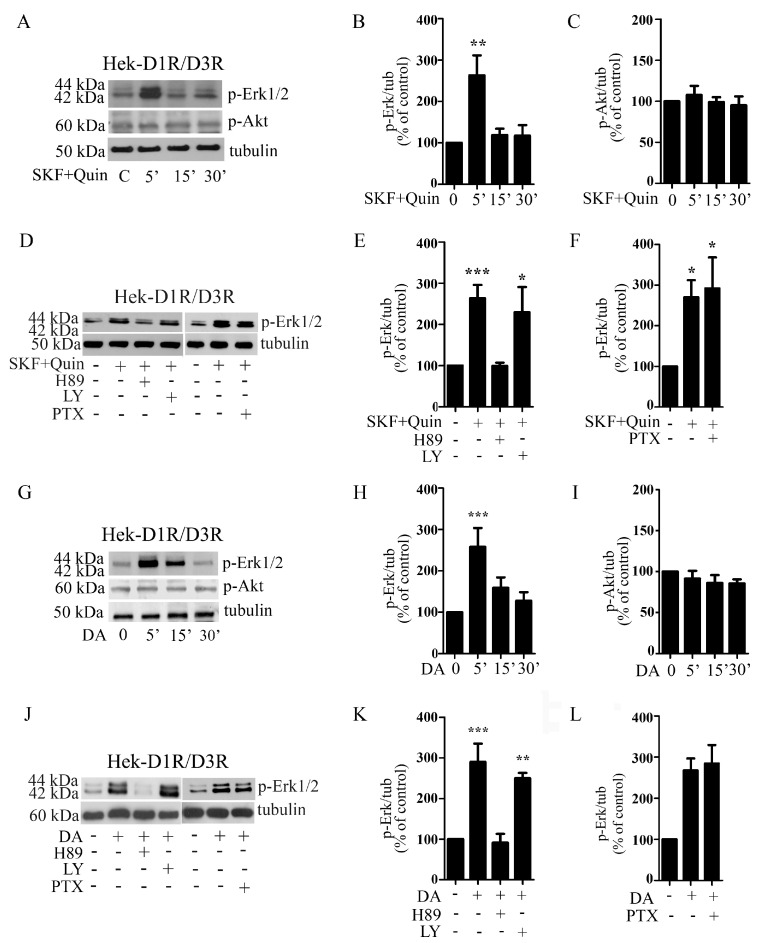
In Hek-D1R/D3R cells, D1R and D3R co-stimulation promotes the activation of the PKA-Erk1/2 pathway. (A) Hek293 cells transiently expressing both the D1R and the D3R (Hek-D1R/D3R) were exposed to both SKF81297 (SKF; 1 μM) and quinpirole (Quin; 1 μM) for 5–30 min (min) and analyzed for Erk1/2 and Akt phosphorylation (p-Erk1/2 and p-Akt, respectively) using Western Blot (WB); a representative WB is shown; (**B**,**C**) densitometric analysis of blots (n = 4) with specific p-Erk1/2 (panel **B**) and p-Akt (panel **C**) levels normalized to the corresponding tubulin levels. Bars represent the mean ± S.E.M. *** *p* < 0.001 vs. untreated cells. (**D**) Hek-D1R/D3R cells were exposed to both SKF (1 μM) and Quin (1 μM) for 5 min in the presence or in the absence of H89 (1 μM) or LY294002 (LY, 10 μM) added 30 min before agonists challenging. PTX (100 ng/mL) was administered overnight before agonists treatment. Cells were analyzed for p-Erk1/2; representative WB are shown; (**E**,**F**) densitometric analysis of blots with specific p-Erk 1/2 levels normalized to the corresponding tubulin levels. Bars represent the mean ± S.E.M. *** *p* < 0.001 vs. untreated cells; (n = 4) (**G**) Hek-D1R/D3R cells were exposed to DA (1 μM) for 5–30 min and analyzed for p-Erk1/2 and p-Akt using WB; a representative WB is shown; (**H**,**I**) densitometric analysis of blots (n = 4) with specific p-Erk1/2 (panel **H**) and p-Akt (panel **I**) levels normalized to the corresponding tubulin levels. Bars represent the mean ± S.E.M. *** *p* < 0.001 vs untreated cells. (**J**) Hek-D1R/D3R cells were exposed to either H89 (1 μM) or LY294002 (LY, 10 μM) for 30 min and challenged with DA (1 μM) for 5 min. In another set of experiments, cells were treated overnight with PTX (100 ng/mL), followed by exposure to DA (1 μM) for 5 min, and analyzed for p-Erk1/2; representative WB are shown; (**K**,**L**) densitometric analysis of blots (n = 4) with specific p-Erk1/2 levels normalized to the corresponding tubulin levels. Bars represent the mean ± S.E.M. *** *p* < 0.001, ** *p* < 0.01, * *p* < 0.05 vs. untreated cells. Data were statistically analyzed with one-way ANOVA followed by post hoc comparison with the Bonferroni test.

**Figure 3 biomolecules-13-00473-f003:**
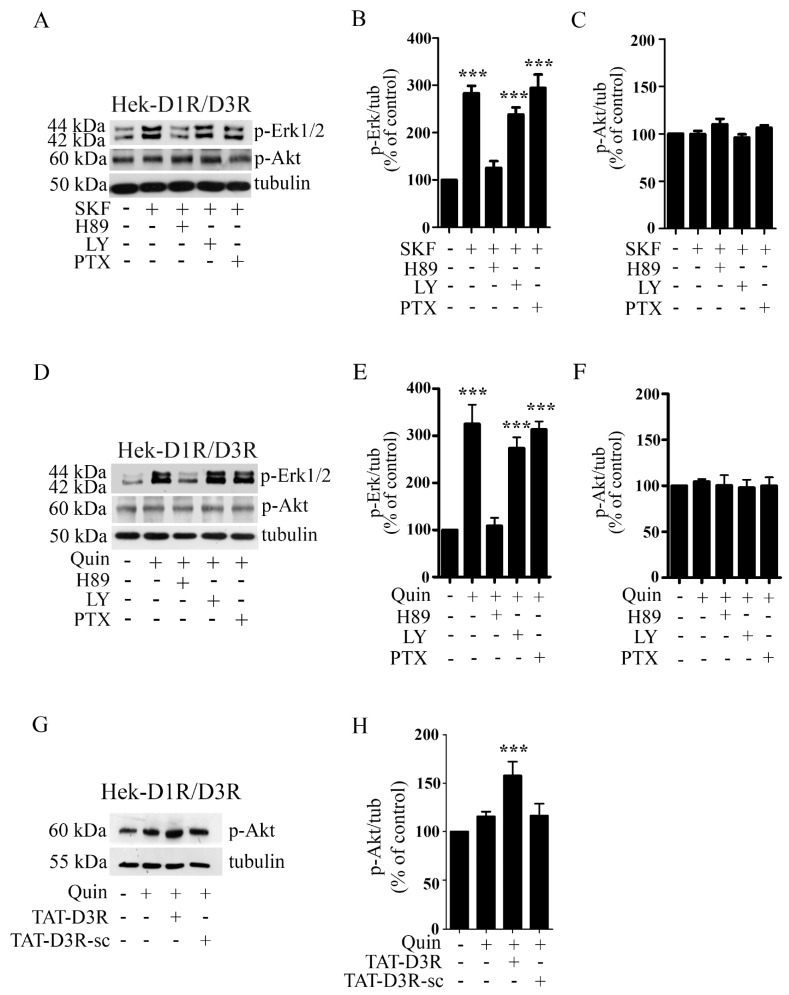
In Hek-D1R/D3R cells, D1R and D3R individual stimulation promotes the activation of the PKA-Erk1/2 pathway. (**A**) Hek-D1R/D3R cells were exposed to SKF (1 μM; 5 min) in the presence or in the absence of H89 (1 μM) or LY294002 (LY, 10 μM) added 30 min before SKF challenging; PTX (100 ng/mL) was administered overnight before SKF treatment. Cells were analyzed for p-Erk1/2 and p-Akt activation using WB; a representative WB is shown; (**B**,**C**) densitometric analysis of blots with specific p-Erk1/2 (panel **B**) and p-Akt (panel **C**) levels normalized to the corresponding tubulin levels. Bars represent the mean ± S.E.M. *** *p* < 0.001, vs. untreated cells; (n = 4). (**D**) Hek D1R/D3R cells were exposed to Quin (1 μM; 5 min) in the presence or in the absence of H89 (1 μM) or LY (10 μM) added 30 min before Quin challenging; PTX (100 ng/mL) was administered overnight before Quin treatment. Cells were analyzed for p-Erk1/2 and p-Akt using WB; a representative WB is shown; (**E**,**F**) densitometric analysis of blots) with specific p-Erk1/2 (panel **B**) and p-Akt (panel C) levels normalized to the corresponding tubulin levels. Bars represent the mean ± S.E.M. *** *p* < 0.001 vs. untreated cells (n = 4). (**G**) Hek D1R/D3R cells were exposed to Quin (1 μM; 5 min) in the presence or in the absence of TAT-D3R (1 μM) or TAT-D3R-sc (1 μM) and analyzed for p-Akt using WB; a representative WB is shown; (**H**) densitometric analysis of blots (n = 4) with specific p-Akt levels normalized to the corresponding tubulin levels. Bars represent the mean ± S.E.M. *** *p* < 0.001 vs. untreated cells. Data were statistically analyzed using one-way ANOVA followed by post hoc comparison with the Bonferroni test.

**Figure 4 biomolecules-13-00473-f004:**
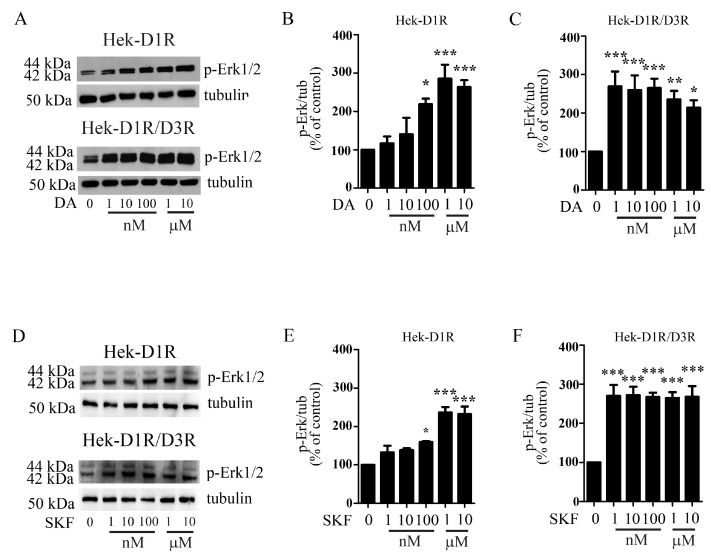
In Hek-D1R/D3R cells, the Interaction of D1R with D3R increases the ability of DA or D1R agonists in activating the Erk1/2 pathway. (**A**) Hek-D1R and Hek-D1R/D3R cells were treated with different doses of DA (1–10 μM) for 5 min and analyzed for p-Erk1/2; a representative WB is shown; (**B**,**C**) densitometric analysis of blots (n = 4) with specific p-Erk1/2 levels normalized to the corresponding tubulin levels. Bars represent the mean ± S.E.M. * *p* < 0.05, *** *p* < 0.001 vs. untreated cells. (**D**) Hek-D1R and Hek-D1R/D3R cells were treated with different doses of SKF (1–10 μM) for 5 min and analyzed for p-Erk1/2; a representative WB is shown; (**E**,**F**) densitometric analysis of blots with specific p-Erk1/2 levels normalized to the corresponding tubulin levels. Bars represent the mean ± S.E.M. * *p* < 0.05, ** *p* < 0.01, *** *p* < 0.001 vs. untreated cells (n = 4). Data were statistically analyzed using one-way ANOVA followed by post hoc comparison with the Bonferroni test.

**Figure 5 biomolecules-13-00473-f005:**
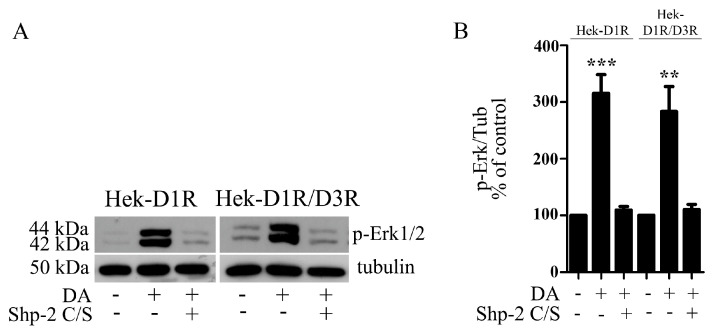
In Hek-D1R/D3R cells, activation of Erk1/2 induced with D1R and D3R co-stimulation requires the tyrosine phosphatase Shp-2. Hek-D1R and Hek-D1R/D3R cells were transiently transfected with a dominant negative form of Shp-2 (Shp-2 (C/S) and treated with DA (1 μM) for 5 min, (**A**) a representative WB is shown; (**B**) densitometric analysis of blots (n = 4) with specific p-Erk1/2 levels normalized to the corresponding tubulin levels. Bars represent the mean ± S.E.M. *** *p* < 0.001, ** *p <* 0.01 vs. untreated cells. Data were statistically analyzed using one-way ANOVA followed by post hoc comparison with the Bonferroni test.

**Figure 6 biomolecules-13-00473-f006:**
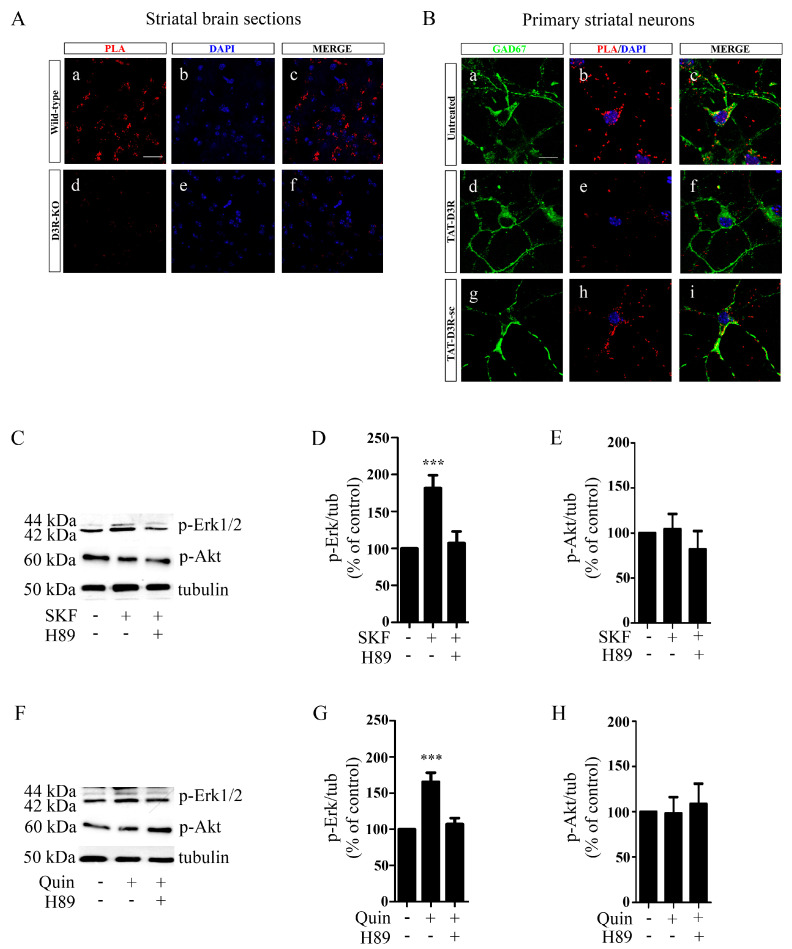
In mouse primary striatal neurons expressing the D1R/D3R heterodimer, D1R and D3R individual stimulation promotes the activation of the PKA-Erk1/2 pathway. ((**A**) Representative PLA image performed in striatal brain sections from Wild-type (**a**–**c**) and D3R-KO mice (**d**–**f**), using antibodies against D1R and D3R; positive PLA signals appear as red spots and nuclei are detected with DAPI. Scale bar: 100 μm. (**B**) Representative PLA image performed in mouse primary striatal neurons untreated (panel **a**–**c**) and pre-treated with TAT-D3R (1 μM) (panel **d**–**f**) or TAT-D3R-sc (1 μM) (panel **g**–**i**) for 24 h; specific anti-D1R and anti-D3R antibodies were used. Striatal neurons were detected using the anti-GAD 67 antibody; PLA signals appear as red spots; nuclei are detected with DAPI (blue). Scale bar = 20 μm. (**C**) Mouse primary striatal neurons were exposed to SKF (10 μM; 5 min) in the presence or in the absence of H89 (1 μM) added 30 min before SKF challenging and analyzed for p-Erk1/2 and p-Akt using WB; a representative WB is shown. (**D**,**E**) Densitometric analysis of blots with specific p-Erk1/2 and p-Akt levels normalized to the corresponding tubulin levels. Bars represent the mean ± S.E.M. *** *p* < 0.001 vs untreated cells (n = 6) (**F**) Mouse primary striatal neurons were exposed to the Quin (10 μM; 5 min) in the presence or in the absence of H89 (1 μM) added 30 min before Quin challenging and analyzed for p-Erk1/2 and p-Akt using WB; representative WBs are shown. (**G**,**H**) Densitometric analysis of blots with specific p-Erk1/2 (**G**) and p-Akt (H) levels normalized to the corresponding tubulin levels. Bars represent the mean ± S.E.M. *** *p* < 0.001 vs. untreated cells (n = 6). Data were statistically analyzed using one-way ANOVA followed by post hoc comparison with the Bonferroni test.

**Figure 7 biomolecules-13-00473-f007:**
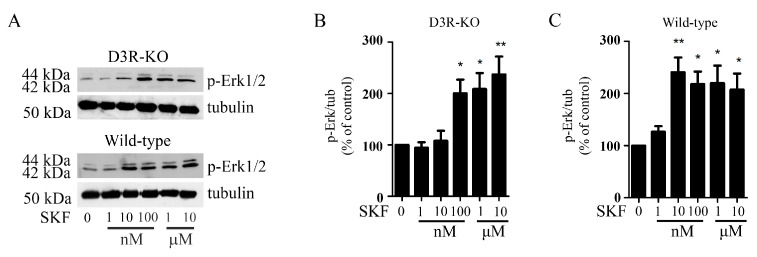
In mouse primary striatal neurons, interaction of D1R with D3R increases the ability of DA or D1R agonists in activating the Erk1/2 pathway. (**A**) Primary striatal neurons from D3R-KO and wild-type mice were exposed to different doses of SKF (1–10 μM; 5 min) and analyzed for p-Erk1/2; representative WBs are shown. (**B**,**C**) Densitometric analysis of blots with specific p-Erk1/2 levels normalized to the corresponding tubulin levels. Bars represent the mean ± S.E.M. * *p* < 0.05, ** *p* < 0.01 vs. untreated cells. (n = 6) (Data were statistically analyzed using one-way ANOVA followed by post hoc comparison with the Bonferroni test).

**Figure 8 biomolecules-13-00473-f008:**
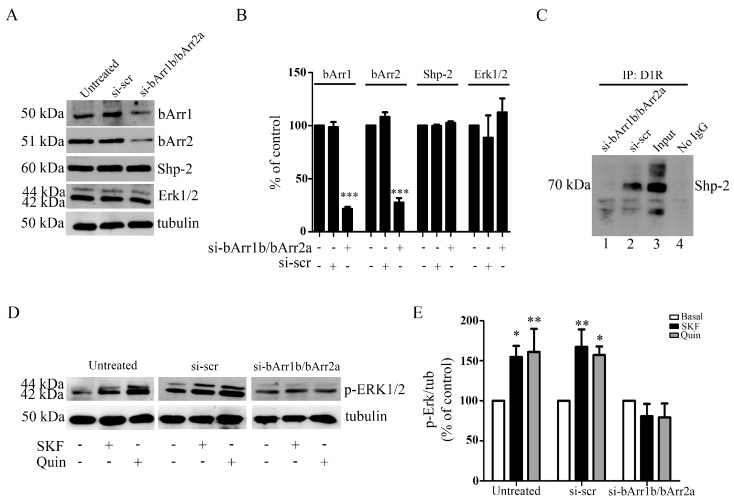
In mouse primary striatal neurons, beta-arrestin 1/2 are required for activation of Erk1/2 induced with D1R and D3R stimulation. (**A**) Neurons were exposed to a combination of two selective siRNA (si-bArr1b and si-bArr2a; 50nM) and to a non-targeting scrambled sequence (si-scr) for 120 h and analyzed for beta-arrestin1 (bArr1), beta-arrestin2 (bArr2), Shp-2 and total Erk1/2 levels by WB; a representative WB is shown; (**B**) Densitometric analysis of blots (n = 3) with specific signals normalized to the corresponding tubulin levels; bars represent the mean ± S.E.M *** *p* < 0.001 versus untreated cells. (**C**) Representative blot of co-immunoprecipitation experiments (IP; n = 3) of D1R and Shp-2 using the anti-D1R antibody in striatal neurons (60 ug for each condition), untreated (lane 3) or treated with si-bArr1b/si-bArr2a (lane 1) or si-scr (lane 2) (both at 50 nM; 120 h); omission of the primary antibody was used as a control (lane 4). (**D**) Untreated, si-scr- and si-bArr1b/si-bArr2a-treated neurons were exposed to SKF (10 μM) or Quin (10 μM) for 5 min and analyzed for p-Erk1/2; representative WBs are shown. (**E**) Densitometric analysis of blots (n = 3) with specific levels of p-Erk1/2 normalized to the corresponding tubulin levels. Bars represent the mean ± S.E.M. * *p* < 0.05, ** *p* < 0.01 vsuntreated cells. Data were statistically analyzed using one-way ANOVA followed by post hoc comparison with the Bonferroni test.

## Data Availability

The raw data used in this study are available on request from the corresponding author.

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
