# Peer review of "G Protein-Dependent Activation of the PKA-Erk1/2 Pathway by the Striatal Dopamine D1/D3 Receptor Heteromer Involves Beta-Arrestin and the Tyrosine Phosphatase Shp-2"

_biomolecules, 2023, doi:10.3390/biom13030473_

Round 1

Author Response

1 and 2: Introduction is so conclusive and needs more concise; The conclusion section needs more concise, and the “discussion” should be put into the “Discussion” section.

Introduction, Discussion and Conclusions have been revised and changed as suggested by the Rev. Please note that only the main changes have been highlighted in the text.

  1. The authors have mentioned that the simultaneous stimulation of D1R and D3R within the heteromer results in the activation of Erk1/2 and Akt, which is conflicted with the results in this study. The authors need to discuss.

As described in the introduction, previous works published by the group of Guitart et al. (2014 and 2019) and performed in transfected cells, have shown that the simultaneous stimulation of D1R and D3R within the heteromer results in the synergistic activation of both Erk1/2 and Akt in a G protein-independent, beta-arrestin 1-dependent way (lines 50-74, track changes version). However, our data, obtained in transfected and corroborated in primary striatal neurons, show that stimulating cells with D1R and D3R agonists within the heterodimer results in the activation of Erk1/2 but not of Akt, with a mechanism that involves both beta-arrestin and the Gs protein. Explaining why the data disagree is for sure an important issue, as suggested by the Rev, to which we have already devoted a large part of the discussion in the first version of the manuscript (lines 600-617, track changes version). Briefly, it is well known that working with heterologous cell systems could originate artifacts according to the different levels of receptor expression. In these systems, in fact, receptors are overexpressed due to the transfection procedures and are rarely under control by the experimenter. In particular, for GPCR heteromers, receptor concentration may deeply impact the formation of a GPCR oligomer unit with multiple possible partners and distinct potential arrangements lean intracellular signaling output outputs. The different D1R and D3R expression levels in transfected cells may explain why our data and those of Guitart et al are not totally superimposable.

4 In Fig 2, the analysis for Erk 1/2 and Akt phosphorylation by Western Blot induced by individual H89, LY and PTX should be conducted to confirm these inhibitors have no effect on the baseline;

We agree with that Rev that showing the inability of H89, LY, and PTX in changing the basal levels of p-Erk1/2 and p-Akt is an important topic. This issue was, in fact, preliminary tested in the group of experiments carried out in Hek cells individually transfected with D1R o with D3R (Supplementary Materials), but partially omitted in the first submission. Therefore, experiments with Hek-D1 and Hek-D3R cells treated with H89 (1 µM) and LY (10 µM) and analysed for p-Akt and p-Erk1/2 have been added as a new panel in Fig S1 (Panels F and G) and S2 (Panels J-L). Please note that PTX inability to change p-Erk1/2 and p-Akt basal levels had been already shown in Fig S2. Changes in the revised version of the Supplementary materials have been highlighted in the text.

 5 In Section 3.2, the experiments in the cell models individually expressing D1R or D3R should be performed, and the signal pathway should be compared with that co-express D1R/D3R.

We certainly agree that experiments with Hek cells individually expressing D1R and D3R are crucial and preliminary for the study of the D1R/D3R heteromer in cells co-expressing both receptors. These experiments had been already performed and shown in the Supplementary Figures of the first submission (S1 and S2). As described above, in the revised form of the Supplementary Data, we also included experiments showing that in Hek-D1R or Hek-D3R cells, H89, LY, and PTX inhibitors did not affect both p-Erk1/2 and p-Akt basal levels.

  1. Please demonstrate how to count the percentage of the cells with positive red spots (Line 287, Page 6).

PLA signal analyses, already partially described in the methods section (lines 244-248, track changes version), have now best explained in the results as suggested by the Rev (lines 272-275, track changes version).

Reviewer 2 Report

I have read the manuscript entitled 'G protein-dependent activation of the PKA-Erk1/2 pathway by the striatal dopamine D1/D3 receptor heteromer involves beta-arrestin and the tyrosine phosphatase Shp-2.’. The authors presented data that bolster our knowledge about the molecular mechanism that can be involved in the activation of the striatal dopamine D1/D3 heteromer. The presented data are very interesting, well described, and discussed. I have only few minor comments for the authors to consider.

In the methods parts, the pharmacological role of used compounds should be explained. In the present form the authors only gave the names of reference compounds e.g. SKF81297, H89, LY294002 without explanating their pharmacological role. These information are given in the results parts, sometimes are repeated, but in my opinion, it should be given in the methods part for all reference compounds (with the full chemical names and salts used), while in the results only the symbols of these compounds should be used. The explanation of the abbreviations used are not given in the first use of them (eg. PLT ll169, RT ll 197) which makes the manuscript hard to read. In the fig 2 title there is a mistake in the (D) and (F) sections - authors wrongly described the WB for p-Akt but there are no data presented.

The manuscript is written in a professional language. However, I think it needs a little adjustment. Sentences are very long and complex. It is worth rewording them and instead of using so many commas or semicolons, divide them into shorter but easier-to-understand sentences.   

Author Response

I have read the manuscript entitled 'G protein-dependent activation of the PKA-Erk1/2 pathway by the striatal dopamine D1/D3 receptor heteromer involves beta-arrestin and the tyrosine phosphatase Shp-2.’. The authors presented data that bolster our knowledge about the molecular mechanism that can be involved in the activation of the striatal dopamine D1/D3 heteromer. The presented data are very interesting, well described, and discussed. I have only few minor comments for the authors to consider.

In the methods parts, the pharmacological role of used compounds should be explained. In the present form the authors only gave the names of reference compounds e.g. SKF81297, H89, LY294002 without explanation their pharmacological role. These information are given in the results parts, sometimes are repeated, but in my opinion, it should be given in the methods part for all reference compounds (with the full chemical names and salts used), while in the results only the symbols of these compounds should be used.

We thank the reviewer for his/her suggestion. Accordingly, all the information about the pharmacological role of each compound as well as their full chemical names has been added in the Materials and Methods section while removed from Results, when appropriate (lines 103-113, track changes version).

The explanation of the abbreviations used are not given in the first use of them (eg. PLT ll 169, RT ll 197) which makes the manuscript hard to read.

The entire manuscript was revised and checked as suggested by the Rev (changes are highlighted).

 In the fig 2 title there is a mistake in the (D) and (F) sections - authors wrongly described the WB for p-Akt but there are no data presented.

We thank the rev for the suggestion, correcting the mistake accordingly.

The manuscript is written in a professional language. However, I think it needs a little adjustment. Sentences are very long and complex. It is worth rewording them and instead of using so many commas or semicolons, divide them into shorter but easier-to-understand sentences.

The entire manuscript has been revised, as suggested by the Rev. Please note that only the main changes have been highlighted in the text.

Reviewer 3 Report

It would be nice if in the conclusions the authors could  also mention that there exists a selectivity. As pointed out by the authors, the D1 and D3 receptors can participate in a large number of hetero receptor complexes besides forming the D1DR-D3R hetero complex. However, they do not form complexes with all types of GPCRs , iono-tropic receptors  and RTKs which should be mentioned. 

Author Response

It would be nice if in the conclusions the authors could also mention that there exists a selectivity. As pointed out by the authors, the D1 and D3 receptors can participate in a large number of hetero receptor complexes besides forming the D1DR-D3R hetero complex. However, they do not form complexes with all types of GPCRs , iono-tropic receptors and RTKs which should be mentioned

We thank the Rev for his/her suggestion. Much evidence was collected since the 80s, when the concept of GPCRs heteromerization emerged, which like suggest that there is no a priori selectivity. GPCRs heterodimerization, in fact, appears as a widespread phenomenon allowing function and regulation of those receptors that are in close proximity within plasma membranes of native tissues. The essential prerequisite for heterodimerization is, therefore, the simultaneous occurrence in the same cellular microdomain, since the high conformational flexibility of GPCRs hypothetically allows the engagement with a great variety of interacting partners (Farran 2017, Pharmacol Res; Guidolin et al., 2019, Frontier in Endocrinol). As described, in the Discussion both D1R and D3R and have been clearly found to interact with other GPCRs and ionotropic receptors. (e.g. D1R with NMDAR and D3R with nAChR). Even if less investigated, GPCRs could also interact with RTK, which represents an additional way of communicating in addition to functional transactivation. Some examples include adenosine receptors with receptors for EGF and FGF, serotonin receptors with FGFR (Di Liberto et al., 2019, Neuropharmacology) or serotonin receptors with TrkB  (lchibaeva et al.,  2022, Cell). This issue is now briefly discussed (lines 681-687, track changes version).

Round 2

Reviewer 1 Report

The authors have alread addressed all my questions. I suggest to accept this paper for publication.